# Antibiotic Treatment Induces Long-Lasting Effects on Gut Microbiota and the Enteric Nervous System in Mice

**DOI:** 10.3390/antibiotics12061000

**Published:** 2023-06-01

**Authors:** Giulia Bernabè, Mahmoud Elsayed Mosaad Shalata, Veronica Zatta, Massimo Bellato, Andrea Porzionato, Ignazio Castagliuolo, Paola Brun

**Affiliations:** 1Department of Molecular Medicine, University of Padova, Via A. Gabelli, 63–35127 Padova, Italy; giulia.bernabe@unipd.it (G.B.); mahmoudelsayedmosaad.shalata@studenti.unipd.it (M.E.M.S.); veronica.zatta@studenti.unipd.it (V.Z.); ignazio.castagliuolo@unipd.it (I.C.); 2Department of Information Engineering, University of Padova, Via G. Gradenigo, 6–35131 Padova, Italy; massimo.bellato@unipd.it; 3Department of Neuroscience, University of Padova, Via A. Gabelli, 61–35127 Padova, Italy; andrea.porzionato@unipd.it; 4Microbiology and Virology Unit of Padua University Hospital, School of Medicine, Via Ospedale, 1–35127 Padova, Italy

**Keywords:** gut microbiota, enteric nervous system, antibiotic, gastrointestinal motility disorder, inflammation

## Abstract

The side effects of antibiotic treatment directly correlate with intestinal dysbiosis. However, a balanced gut microbiota supports the integrity of the enteric nervous system (ENS), which controls gastrointestinal neuromuscular functions. In this study, we investigated the long-term effects of antibiotic-induced microbial dysbiosis on the ENS and the impact of the spontaneous re-establishment of the gut microbiota on gastrointestinal functions. C57BL/6J mice were treated daily for two weeks with antibiotics. After 0–6 weeks of antibiotics wash-out, we determined (a) gut microbiota composition, (b) gastrointestinal motility, (c) integrity of the ENS, (d) neurochemical code, and (e) inflammation. Two weeks of antibiotic treatment significantly altered gut microbial composition; the genera *Clostridium*, *Lachnoclostridium*, and *Akkermansia* did not regain their relative abundance following six weeks of antibiotic discontinuation. Mice treated with antibiotics experienced delayed gastrointestinal transit and altered expression of neuronal markers. The anomalies of the ENS persisted for up to 4 weeks after the antibiotic interruption; the expression of neuronal HuC/D, glial-derived neurotrophic factor (*Gdnf*), and nerve growth factor (*Ngf*) mRNA transcripts did not recover. In this study, we strengthened the idea that antibiotic-induced gastrointestinal dysmotility directly correlates with gut dysbiosis as well as structural and functional damage to the ENS.

## 1. Introduction

The microbiota refers to a group of beneficial and potentially harmful microorganisms that colonize and interact as a community within the host’s specific area [1]. One of these defined spaces is the gastrointestinal tract, wherein a separate organ called the gut microbiome lies. The gut microbiome has been an area of interest for many researchers for its partly discovered physiological or pathological effects, which result from its composition [2]. Many factors, such as diet, demographics, gender, and drugs, affect the richness and diversity of the microbiota composition and can lead to a condition called gut dysbiosis, which can indirectly impact an individual’s health status. Indeed, multiple health conditions have been connected to dysbiosis, including cardiovascular diseases, obesity, and diabetes [2,3]. However, the gut microbiome is not the only factor responsible for maintaining the health of the gastrointestinal tract. Another major player is the enteric nervous system (ENS), a complex neurological network composed of neurons and enteric glial cells that controls many aspects related to gastrointestinal functions, such as motility, secretion, blood flow, hypersensitivity, and immune response. Therefore, any deterioration in the ENS can lead to multiple gastrointestinal disorders [4,5,6,7]. Recently, efforts have been made to draw a connection between the gut microbiome, the ENS, and the underlying mechanisms building this connection [6]. It has been reported that the gut microbiome can affect the ENS’s neurogenesis through the modulation of serotonin production, which stimulates the peristaltic movement of the gut as well [8,9]. Indeed, germ-free mice experience alterations in gut morphology and motility; their cecum is enlarged, and the small intestine is less developed with reduced gut motility partly due to decreased sensory and motility innervation [10,11,12]. Moreover, bacterial endotoxins (i.e., lipopolysaccharide (LPS)) and metabolites (i.e., the short-chain fatty acids, such as butyrate) produced by bacterial fermentation are known to impact gastrointestinal functioning and ENS’s function and structure. Consequently, any influence on the gut microbiome can indirectly influence the ENS [4,13,14].

Antibiotics are considered gut-dysbiosis-promoting agents, which can significantly affect the diversity and richness of the microbiome. Antibiotics have short- and long-term effects on the microbiome, and the recovery from these effects depends on many factors, such as the antibiotic spectrum of activity, dose, duration of treatment, and the combination of antibiotics [3,15]. The recovery from antibiotic use is usually characterized by a slow and gradual retrieval of the same diversity and richness achieved prior to antibiotic use. Palleja et al. [16] reported a rise in the number of harmful Enterobacteriaceae and a decline in beneficial bacteria such as *Bifidobacterium* and species producing short-chain fatty acids in 12 healthy young adult males treated with a cocktail of antibiotics. The antibiotic-induced effects appeared to be evident up to six months after antibiotic discontinuation.

The reduction in richness and the change in the composition of the microbiome might indirectly affect both the ENS and gastrointestinal functions. It has been reported that mice treated with antibiotics suffered drastic changes in immune cell populations, an enlarged cecum, and narrower villi. The structural alterations are accompanied by functional alterations in the transit time along the whole ileum [17]. Moreover, the administration of vancomycin to neonatal mice changes the composition of the gut microbiome and the development and differentiation of myenteric neurons, as evidenced by the reduction in neuronal markers such as the pan-neuronal marker HuC/D and neuronal nitric oxide synthase (nNOS) in young animals [18,19]. In this study, we investigated the long-term (up to six weeks) effects of antibiotic-induced microbial dysbiosis on the structure and functions of the ENS and analysed the impact of the spontaneous re-establishment of the gut microbiota on gastrointestinal functions. We observed that gut microbiota composition was not completely re-established in the cecum of mice even after six weeks of antibiotic discontinuation. At the same time, the antibiotic-induced ENS alterations persisted for up to 4 weeks after the antibiotic interruption. Our findings highlight the bacterial genera of the gut microbiota that slowly recover after antibiotic treatment. Therefore, supplementation with selected prebiotics could foster the recovery of gut microbiota composition and ENS functions after antibiotic therapy.

## 2. Results

### 2.1. Antibiotic Treatment Causes Long-Lasting Alterations in Gut Microbiota

Antibiotic-induced intestinal microbiota alterations were initially assessed on total DNA isolated from cecal content by estimating 16S rRNA abundance using primers for the V2 and V6 regions of bacterial 16S genes [20]. As reported in Figure 1, mice treated with antibiotics had a significantly reduced bacterial load compared to vehicle-treated mice (*p* < 0.02) for both the V2 and V6 regions. The severe reduction in copy number in the bacterial genome persisted for two weeks after antibiotic discontinuation (2 weeks w/o), and recovered starting from the fourth week of wash-out (Figure 1a). We determined a significant reduction in the operational taxonomic units (OTUs, Figure 2b,c) due to antibiotic treatment (*p* < 0.05 vs. vehicle). Interestingly, the OTU relative abundance shrinkage persisted for two weeks after antibiotic discontinuation (Figure 2b). By analysing the differences in beta diversity among the experimental groups (Figure 2c), we observed that samples obtained from vehicle-treated mice clustered significantly differently than samples from antibiotic-treated animals and samples obtained from mice sacrificed two weeks after antibiotic withdrawal (*p* = 0.002 vs. vehicle-treated group). Samples from antibiotic-treated mice sacrificed immediately after the treatment or two weeks after antimicrobial discontinuation displayed greater dissimilarity along PC2 (Figure 2c). At the same time, samples from mice sacrificed four and six weeks after antibiotic discontinuation reported more significant similarity in taxonomical characterization compared with samples obtained from mice treated with antibiotics for two weeks and then sacrificed (Figure 2c).

Genera abundances revealed differences in antibiotic-treated and vehicle-treated mice (Figure 1d). Indeed, antibiotics depleted genera of bacteria, namely *Porphyromonas*, *Clostridium*, *Akkermansia*, and *Bacteroides*; *Lactococcus* was the dominant genus in the antibiotic-treated group. During antibiotic discontinuation, the prevalence of *Lactococcus* gradually reduced, and the prevalence of *Bacteroides* increased from 2 to 6 weeks post wash-out. However, in the sixth week after antibiotic wash-out, the cecal microbiota did not resume its original composition; the genera *Clostridium*, *Lachnoclostridium*, and *Akkermansia* did not regain the relative abundance described before antibiotic administration (Figure 1d). Therefore, antibiotic-induced gut dysbiosis persisted for up to 6 weeks after antibiotic withdrawal.

### 2.2. Antibiotic Treatment Induces Persistent Gut Dysmotility

Although the administration of the antibiotic mixture did not result in diarrhoea, behavioural alterations, or histologic damage in the ileum and colon (Appendix A) at any experimental time point, we observed intestinal dysmotility in treated mice.

While gastric emptying did not significantly change in treated animals (Figure 2a), for mice sacrificed immediately after two weeks of antibiotic administration, severe delays in the progression of fluorescein isothiocyanate–dextran in the ileum were observed compared with mice treated with vehicle (*p* < 0.05), indicating impaired gastrointestinal motility. On the contrary, we observed reduced transit time two and four weeks after antibiotic discontinuation (*p* < 0.05 vs. vehicle). Ileal motility alterations disappeared after six weeks of the antibiotic wash-out (Figure 2b).

As regards colonic motility, antibiotic treatment reduced the frequency of faecal pellets expulsed over 1 h (Figure 2c) and prolonged the retention time of the rectal beads (Figure 2d), indicating delayed colonic movements and impaired colonic mucosa secretion and liquid retention [21]. Colonic dysmotility persisted during antibiotic discontinuation. Indeed, stool pellet frequency increased on the second and fourth weeks of wash-out, whereas the retention time of the beads normalized to control levels after two weeks of wash-out. The frequency of faecal pellets barely returned to the control values after six weeks of antibiotic discontinuation.

### 2.3. Antibiotic Administration Induces Structural Alterations in the Enteric Nervous System

Given that the enteric nervous system (ENS) finely regulates most of the gastrointestinal activities and the gut microbiota preserves the integrity of the ENS [5], we evaluated the impact of antibiotic administration on the architecture of the ENS.

Confocal immunofluorescence performed on ileal whole-mount preparations revealed that expression and distribution of βIII-tubulin, a neuronal marker, were altered in neurons of the myenteric plexus obtained from mice treated with antibiotics for 2 weeks. As reported in Figure 3a, βIII-tubulin aggregated inside the neurons and lost the elongated distribution that is evident in vehicle-treated animals. Structural disorganization of βIII-tubulin persisted after four weeks of antibiotic discontinuation and barely recovered in mice following six weeks of antibiotic wash-out (Figure 3a). Moreover, the expression of HuC/D, a critical neuronal regulatory protein, significantly decreased after antibiotic administration, as determined using Western blot analysis and the relative densitometric analysis (Figure 3b,c). S100β, a Ca^2+^-modulated protein of the enteric glial cells, was also diminished 3-fold following antibiotic treatment (Figure 3b,c). Following antibiotic discontinuation, we observed a plasticity in the expression of neuronal and glial proteins, which tended to regain the values reported before antibiotic administration in the second week. However, altered expression persisted for up to six weeks for HuC/D and four weeks for S100β (Figure 3c).

### 2.4. Antibiotic Alters the Neurochemical Code in the Enteric Nervous System

In addition to the structural anomalies in the ENS, the immunofluorescence analysis performed on ileal whole-mount preparations demonstrated that antibiotic administration caused a drastic reduction in Substance P immunoreactivity until the fourth week after antibiotic discontinuation (Figure 4a). At the sixth week of antibiotic withdrawal, expression of Substance P was comparable with that in vehicle-treated animals. Moreover, the expression of nNOS was significantly reduced in antibiotic-treated mice both sacrificed after antibiotic administration and following two weeks of antibiotic discontinuation (Figure 4b,c). At four and six weeks of antimicrobial discontinuation, the expression of nNOS regained the values reported in vehicle-treated mice (Figure 4b,c).

Through qPCR performed on LMMP preparations, we found a significant decrement in mRNA transcripts specific for the glial-derived neurotrophic factor (*Gdnf*), nerve growth factor (*Ngf*), ciliary neurotrophic factor (*Cntf*), and neurotrophins (*Ntf*) 5 and 3 (Figure 5a–e). Specific mRNA transcript levels for the brain-derived neurotrophic factor (*Bdnf*, Figure 5f) and leukaemia inhibitory factor (Lif, not shown) were comparable to those of vehicle-treated mice. Considering the above-reported data on alterations in gut microbial composition (Figure 1) and derangement in the ENS architecture (Figure 3 and Figure 4), we infer that the neurochemical code of enteric nerves matches the damage to the ENS. Moreover, the glial-derived neurotrophic factor is more susceptible to antibiotic-induced alterations than the neuronal-related factors.

### 2.5. Antibiotic Administration Triggers Inflammation in the Myenteric Plexus

We next evaluated whether antibiotic treatment triggers inflammation in the myenteric plexus. Even though we have no evidence of an inflammatory infiltrate (Appendix A), TNF-α levels significantly increased after antibiotic treatment (Figure 6a); higher levels of the pro-inflammatory cytokine were also evident in the LMMP obtained from mice during the second and fourth weeks of antibiotic discontinuation. In addition, IFN-γ and IL-1β significantly increased in the LMMP following antibiotic administration (Figure 6b,c); these cytokines regained the values observed in vehicle-treated mice starting the second week after antibiotic discontinuation, leading us to believe that the inflammatory milieu is not due to gut dysbiosis but a consequence of the ENS alterations.

## 3. Discussion

The incidence of side effects of antibiotic therapy is not trivial since 5–30% of subjects taking antibiotics will suffer from diarrhoea for 2–6 days [22]. Diarrhoea has been generally viewed as a consequence of gut microbiota disruption, favouring the overgrowth of pathogenic bacteria such as *C. difficile* [23]. However, the identification of *C. difficile* as the etiologic agent occurs in less than 10% of episodes of antibiotic-associated diarrhoea [24]. Moreover, the high frequency of gastrointestinal symptoms other than diarrhoea suggests that other mechanisms might be involved. In this study, we provided evidence that, at the genera level, antibiotic-induced dysbiosis persists for up to six weeks after antibiotic discontinuation. The alterations in gut microbiota composition parallel intestinal dysmotility and structural anomalies of the enteric nervous system (ENS). In this study, we observed that antibiotic administration quickly diminished bacterial diversity of the cecal content, resulting in increased abundance of the genera *Lactococcus*, and antibiotic discontinuation led to a partial recovery in microbiota composition (Figure 1d). Our findings are in accordance with several experimental and clinical studies reporting selective genera depletion following antibiotic administration [15,25]. In this study, we observed that genera such as *Clostridium*, *Lachnoclostridium*, and *Akkermansia* did not completely regain the relative abundance observed before antibiotic treatment (Figure 1d). Moreover, the depletion of these bacterial genera is consistent with the severity of structural and functional alterations in the ENS (Figure 2 and Figure 3). Of course, in our study, we could not exclude the direct toxic effects of the antibiotic mixture on neurons. Indeed, several classes of antibiotics can directly affect muscle and neuronal cells [26]. However, data regarding this issue are contradictory. In guinea pigs’ small intestine, erythromycin inhibits nerve-mediated longitudinal muscle contractions by acting on enteric nerves [27]. On the contrary, tetracycline directly protects neurons in an ischemia-like injury in vitro model [28]. Transcriptomic analysis revealed enrichment in signalling pathways associated with neurotransmitters such as dopamine and norepinephrine following the administration of polymyxin B [29].

Recovery of intestinal bacterial diversity following antibiotic therapy is described in most individuals, although effects on the gut community can persist for several weeks, mainly following the administration of antibiotics with strong and broad activity against anaerobes [30,31]. The restored gut microbiota, however, incompletely resembles the pre-treatment state [30]. In this study, we found that the severe antibiotic-induced changes in the gut microbiota occurred in parallel with the functional anomalies of the gastrointestinal tract. Indeed, the gut microbiota is critical for maintaining intestinal homeostasis since changes in bacterial composition are involved in the pathophysiology of functional and inflammatory bowel disorders. The involved cellular populations and the underlying molecular mechanisms are only partially elucidated [32,33]. Thus, components of the healthy intestinal microbiota interact with specific sensors of the innate immune and entero-endocrine systems as well as cells of the ENS to generate cell signals that are crucial for guaranteeing epithelial barrier integrity, instructing the local immune responses, and supporting the functional integrity of myenteric neurons [5,34]. We believe that the delay in recovering gut microbiota composition following antibiotic discontinuation observed in our study explains the persistence of the ENS alterations. As reported in the literature, commensal gut bacteria modulate synaptic proteins, neurotransmitter systems, brain development, and behaviour in infant development [35]. In vitro co-culture models indicated that the communication between probiotic strains such as *Lactobacillus rhamnosus* or *Lactobacillus fermentum* and neuronal cells through the epithelial barrier is mandatory for neurite outgrowth [36]. Finally, oral *Lactobacillus* spp. administration leads to an increase in brain neurotransmitters that persists for up to 4 weeks after treatment has ceased [37]. On the contrary, impairments in the gut microbial community alter the production of critical trophic signals for neurons, such as short-chain fatty acids, paving the way for neuromuscular dysfunctions [4,13]. Oral administration of antimicrobials to germ-free mice did not affect the gastrointestinal neurotransmitters observed in specific pathogen-free animals [38]. Specifically, in colonized mice, vancomycin administration disrupts the neurochemical code of the enteric circuitry and reduces cholinergic, calbindin, and nitric oxide synthase-positive myenteric neurons [39].

Gut dysbiosis is frequently reported in diseases involving inflammation of the gut, including inflammatory bowel disease [32]. In this study, we found that antibiotic-induced gut dysbiosis correlated with increased pro-inflammatory cytokine levels in the myenteric plexus (Figure 6). Even if we did not recognize bacteria endowed with inflammatory activity among the most prevalent genera identified in the microbiota analysis (Figure 1d), we could speculate that the inflammation of the ENS is an indirect effect of the dysbiosis dictated by the alterations in the neurochemical code (Figure 4 and Figure 5). Indeed, neurotransmitters and neuropeptides are emerging as regulators of the inflammatory process in the gut, acting on the local immune cells. Activation of the intestinal immune system generates bidirectional communication between the ENS and the enteric immune system that tunes the magnitude of inflammatory reactions [40].

In conclusion, this study shows that antibiotic administration triggers gastrointestinal anomalies and alters enteric neurons, probably disrupting the balanced availability of trophic signals generated by the gut microbiota [33]. The mixture of antibiotics selectively affects specific bacteria genera, disrupting crucial interactions with the host, such as the activation of innate immune cognate receptors or the availability of specific bacterial metabolites. Further studies are necessary to verify whether supplementation with specific bacteria or bacterial metabolites may favour the structural and functional recovery of the ENS.

## 4. Materials and Methods

### 4.1. Mice Treatment

Six-week-old male C57BL/6J mice were purchased from Charles River Laboratories (Calco, LC; Italy) and housed in a temperature- and humidity-controlled room under a 12 h light/dark cycle. Mice were housed in the same room and maintained by the same personnel. All animals were specifically pathogen-free and given a standard chow diet and tap water ad libitum.

Freshly prepared antibiotic solutions (Abx) consisting of 50 mg/kg vancomycin, 100 mg/kg neomycin, 100 mg/kg metronidazole, and 100 mg/kg ampicillin (all purchased from Merck, Milan, Italy) were administered (100 µL/mouse) every 12 h for 14 days by oral gavage using a stainless-steel feeding tube [5,20]. Control animals received the same volume of the vehicle for the same period. Animals were sacrificed at the end of the antibiotic treatment or 2, 4, or 6 weeks post antibiotic discontinuation (wash-out; Figure 7). Signs of illness, diarrhoea, and spontaneous activity in the cage were monitored during the antibiotic treatment and the wash-out period.

The experimental protocols were approved by the Animal Care and Use Ethics Committee of the University of Padova under license from the Italian Ministry of Health, and they were found to be in compliance with national and European guidelines for the handling and use of experimental animals.

### 4.2. Microbiota Analysis

Cecal contents were collected at the specified time points (see the asterisks in Figure 7) in the early morning hours (between 8 and 10 a.m.) using sterile toothpicks and were placed into sterile 1.5 mL tubes. Samples were stored at −80 °C until analysis. DNA was isolated from 200 mg faecal samples using QIAamp DNA Stool Mini Kit (Qiagen, Milan, Italy) following the manufacturer’s instructions. The purified DNA was quantified via spectrophotometry at 260 nm and stored at −20 °C until PCR amplification. Total bacterial load was quantified based on the 16S rRNA abundance using degenerate primers for the V2 and V6 regions (Table 1) on 50 ng of DNA using SYBR Green Master Mix (Applied Biosystems) in the ABI PRISM 7700 Sequence Detection System (Applied Biosystem) [20]. The number of 16S rRNA gene copies was normalized against the faecal weight (mg) of each sample.

Barcoded libraries of the bacterial 16S rRNA gene hypervariable regions were prepared using faecal DNA extracts. Sequencing was performed using Illumina MiSeq chemistry at the BMR Genomics SRL (Padua, Italy). Amplicons were generated using the modified universal bacterial primer pairs 515F (5′-TCGTCGGCAGCGTCAGATGTGTATAAGAGCAGGTGCCAGCMGCCGCGGTAA-3′) and 806R (5′-GTCTCGTGGGCTCGGAGATGTGTATAAGAGACAGGGAACHVGGGTWTCTAAT-3′) according to the Illumina MiSeq 16S Metagenomic Sequencing Library Preparation protocol [41]. Bioinformatic analyses were conducted using the Quantitative Insights Into Microbial Ecology (QIIME) software (v2.2019.4). Denoising was performed using the DADa2 pipeline and Qiime 2. Taxonomic classification of amplicon sequence variants was performed against the BLASTn reference database. Through alignment with the 2013 release of the Greengenes reference database, operational taxonomic units (OTUs) were clustered at 97% sequence similarity.

### 4.3. Gastrointestinal Transit

Fluorescein isothiocyanate–dextran (70,000 Da molecular weight) dissolved in phosphate-buffered saline (PBS, 6.25 mg/mL) was administered in a total volume of 200 μL by intragastric gavage [5]. Animals were placed in their original cages and then sacrificed after 60 min. The abdomen was opened, and the gut was carefully removed, avoiding stretching. The stomach was examined separately; the small intestine was divided into 8 identical segments. Luminal contents were collected and clarified through centrifugation (10,000× *g*, 15 min, 4 °C). Fluorescence analysis was performed at 494/521 nm (Hitachi F2000; Hitachi, Tokyo, Japan). Data are expressed as a percentage of fluorescence per segment. Gastrointestinal transit was calculated as the geometric centre of distribution of the fluorescent probe [5]. We performed gastrointestinal transit assays between 8 and 10 a.m. to ensure reproducibility among the different time points.

### 4.4. Colonic Motility

Mice fasted for 16 h were lightly anaesthetized with isoflurane (Merial, France), and a 2 mm glass bead was inserted into the distal colon at 2 cm from the anus. Mice readily regained consciousness and were placed individually in a cage. Distal colonic motility was determined by monitoring the time required for the expulsion of the glass bead, reported as the time (in seconds) of bead retention [7].

In a separate set of experiments, non-anaesthetized mice were placed in clean cages; the number of pellets expelled over an hour was determined. Colonic motility assays were always performed in the morning hours (between 8 and 10 a.m.) to ensure reproducibility among the experiments at the different time points.

### 4.5. Histopathological Evaluation

Ileal and colon sections (5 µm thick) were cut from formalin-fixed and paraffin-embedded full-thickness specimens. Slides (6–8 per mouse) were subjected to standard haematoxylin and eosin (H&E) staining for routine histological examination. A minimum of 10 independent fields per animal were examined using a Leica microscope equipped with a digital camera.

### 4.6. Whole-Mount Staining

Segments of distal ileum (4–6 cm long) were flashed with PBS, filled with fixative solution (neutral buffered formalin solution, 10%) and immersed in the same fixative solution [5]. After 1 h, tissues were washed in PBS (3 × 10 min) and stored at 4 °C in PBS. Under a dissecting microscope, a small incision was made with tweezers on a segment of the ileum (approximately 1 cm long) and the longitudinal muscle layer with the adherent myenteric plexus (LMMP) was peeled off. The tissue sheet was gently stretched and pinned down on wax support, washed twice with PBS, incubated in permeabilization buffer (0.5% Triton-X100 in PBS), and then stained at room temperature for 16 h with the proper primary antibody (Table 2). Samples were extensively washed and probed with the appropriate fluorescent-labelled secondary antibodies (Table 2). Tissues were mounted with Prolong Antifade kit (Life Technologies), and samples were imaged using a Leica TCSNT/SP2 confocal microscope. All microscope settings were set to collect images below saturation and were kept constant for all images.

### 4.7. Immunoblot Analysis

At the time of sacrifice, the small intestine was removed, washed in ice-cold PBS, cut into 1 cm long pieces, and placed over a sterile glass rod. The LMMP was quickly peeled off under a dissecting microscope (Leica). LMMPs were homogenized in RIPA buffer (150 mM NaCl, 50 mM Tris-HCl, 0.25% sodium deoxycholate, 0.1% Nonidet P-40, 100 μM NaVO_4_, 1 mM NaF) containing protease inhibitors (0.5 mM EDTA, 0.1 mM PMSF, 1 μM leupeptin, 150 nM aprotinin). After 30 min at 4 °C, particulate material was removed through centrifugation (15,000× *g* for 30 min at 4 °C) and protein concentration in the supernatants was determined using the bicinchoninic acid method (Pierce). Proteins were fractionated through an SDS-PAGE gel, immobilized onto a nitrocellulose membrane, and subsequently subjected to immunoblot analysis. The membranes were blocked for 1 h with 5% non-fat dry milk in PBS with 0.05% Tween 20. Membranes were probed with appropriate primary antibodies (Table 2). Immuno-complexes were detected using horseradish peroxidase (HRP)-conjugated secondary antibodies (Table 2) and an enhanced chemiluminescent system (Millipore). Images were captured using Hyper Film MP (GE Healthcare). The membranes were probed with an anti-mouse β-actin antibody (Merck) to ensure equal loading of the samples. Densitometry analysis of the band intensity was performed using the ImageJ software (US National Institutes of Health; Madison, Wisconsin).

### 4.8. Inflammatory Cytokine Profile

LMMPs were homogenized in phosphate-buffered saline (PBS) (1:10 wt/vol) supplemented with protease inhibitors (1 mmol/L phenylmethylsulfonyl fluoride, 10 μg/mL aprotinin, and 10 μg/mL leupeptin). The samples were centrifuged (10,000× *g* for 10 min at 4 °C), and interleukin (IL)-1β, interferon (IFN)-γ, and tumour necrosis factor (TNF)-α were quantified using an enzyme-linked immunosorbent assay (Biosource, Milan, Italy). Data were normalized by total protein content measured using the bicinchoninic acid method.

### 4.9. RNA Isolation and Quantitative RT-PCR

Total RNA was extracted from LMMP samples using the EZNA lysis buffer (Total RNA Kit I, Omega Bio-tek, Italy). Contaminating DNA was removed through DNase I treatment (Omega Bio-tek). Gene expression was assessed using the iTaq Universal SYBR Green One-Step Kit (Bio-Rad Laboratories; Segrate, Italy). The expression of the targeted mRNA was normalized to 18S ribosomal RNA (*Rn18S*) and plotted as mean fold expression. Oligonucleotides and annealing temperatures used for qRT-PCR are listed in Table 1.

### 4.10. Statistical Analysis

Data are reported as the mean ± standard deviation of the mean (SD) except for gastric emptying and geometric centre, which are reported as median ± SD. One-way analysis of variance followed by the Bonferroni multicomparison test was used to compare the data of three or more groups. The statistical analysis was performed using GraphPad Prism 3.03 software (GraphPad, San Diego, CA, USA). *p* values < 0.05 were considered statistically significant.

## Figures and Tables

**Figure 1 antibiotics-12-01000-f001:**
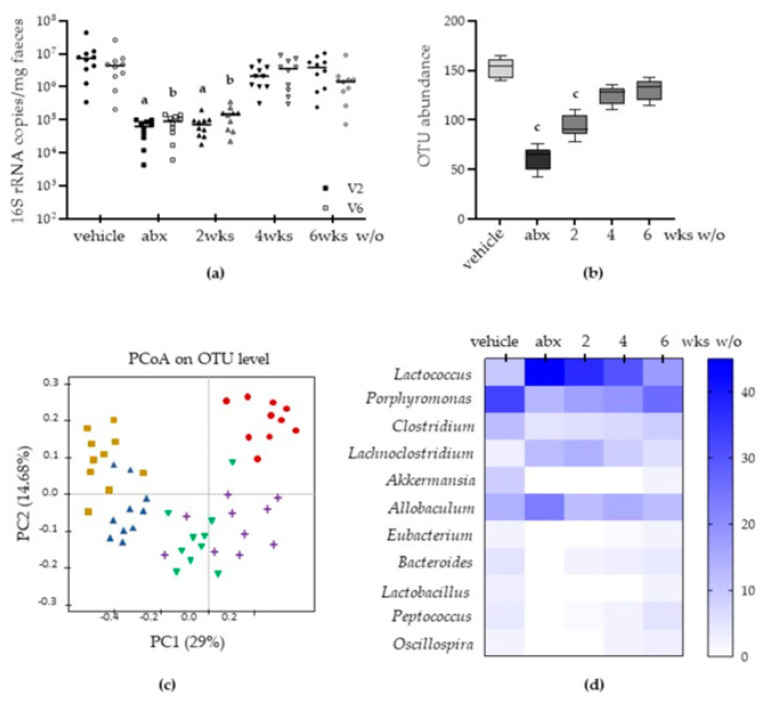
Mice were treated with vehicle or antibiotic mixture (abx) and sacrificed 2 weeks (wks) later; otherwise, mice were treated with abx and sacrificed 2, 4, or 6 weeks (wks) post antibiotic wash-out (w/o). The total bacterial load was estimated from the DNA isolated from the cecal content with qPCR using primers for the V2 and V6 regions of bacterial 16S genes. Dark circle: V2, vehicle; grey circle: V6 vehicle; dark square: V2 abx; grey square: V6 abx; dark up-triangle: V2 2 wks wash-out; grey up-triangle: V6 2 wks wash-out; dark down-triangle: V2 4wks wash-out; grey down-triangle: V6 4 wks wash-out; dark rhombus: V2 6 wks wash-out; grey rhombus: V6 6 wks wash-out (**a**). DNA was sequenced using Illumina MiSeq chemistry; operational taxonomic unit (OTU) relative abundance is reported. Light grey: vehicle; dark grey: abx treatement; intermediate grey: wash-out. (**b**). Taxonomical differences among samples were determined using principal coordinate analysis (PCoA) (**c**). The average relative abundance percentage for the 11 most abundant genera is displayed (**d**). Data are reported as mean ± SD; number of mice for each experimental group = 10 mice per group; ^a^ denotes *p* < 0.02 vs. V2 copy number in vehicle-treated mice; ^b^ denotes *p* < 0.02 vs. V6 copy number in vehicle-treated mice; ^c^ denotes *p* < 0.05 vs. vehicle-treated mice.

**Figure 2 antibiotics-12-01000-f002:**
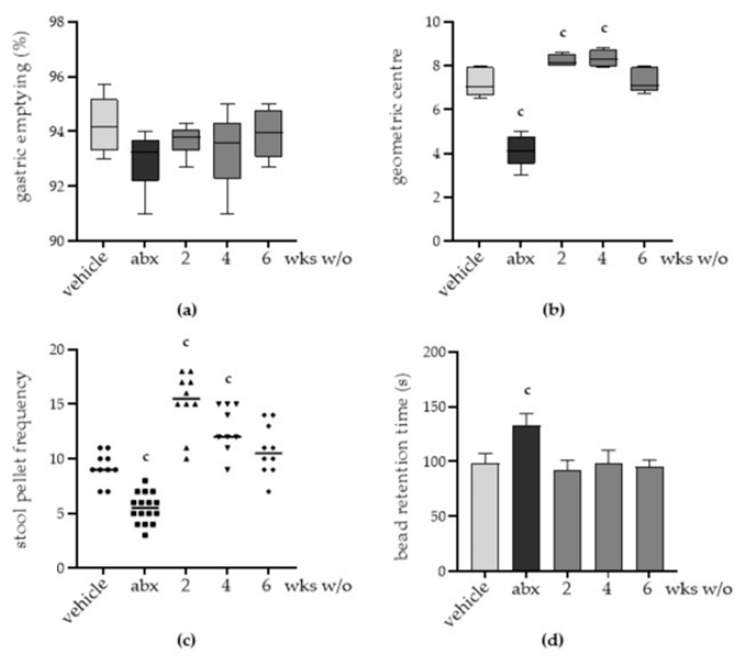
Mice were treated with vehicle or antibiotic mixture (abx) and sacrificed 2 weeks (wks) later; otherwise, mice were treated with abx and sacrificed 2, 4, or 6 weeks (wks) post antibiotic wash-out (w/o). Gastrointestinal dysmotility was determined by administering the animals with non-absorbable FITC-labelled dextran by oral gavage. Animals were sacrificed 60 min later. Gastric emptying was calculated as the percentage of dextran retained in the stomach with respect to the total amount of fluorescence in the gastrointestinal tract. Light grey: vehicle; dark grey: abx; intermediate grey: wash-out. (**a**). The transit of the fluorescent probe in the ileum is reported as the calculated geometric centre, which is the centre of the distribution of fluorescent dextran in the ileum (**b**). The number of faecal pellets expelled in 1 h. Circle: vehicle treated mice; square: abx; up-triangle: 2 wks wash-out; down-triangle: 4 wks wash-out; rhombus: 6 wks wash-out. (**c**). Time (seconds, s) required for the expulsion of a glass bead inserted into the rectum (**d**). Data are reported as mean ± SD. number of mice for each experimental group = 9–16 mice per group. ^c^ denotes *p* < 0.05 vs. vehicle-treated mice.

**Figure 3 antibiotics-12-01000-f003:**
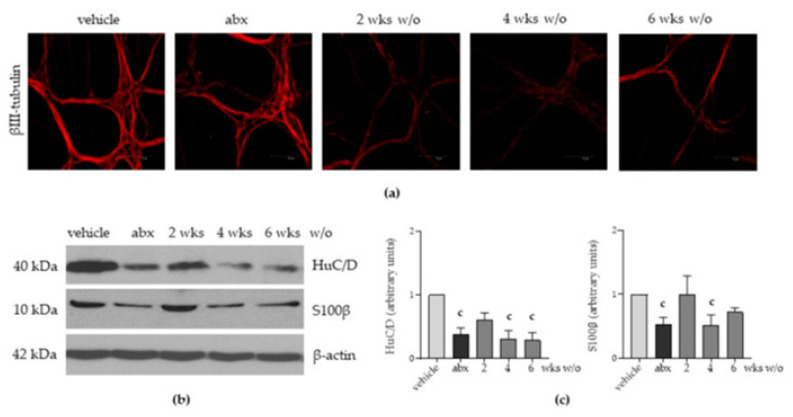
Mice were treated with vehicle or antibiotic mixture (abx) and sacrificed 2 weeks (wks) later; otherwise, mice were treated with abx and sacrificed 2, 4, or 6 weeks (wks) post antibiotic wash-out (w/o). Immunofluorescence on whole-mount preparations of the distal ileum for βIII-tubulin (neuronal markers) was performed. Representative images of 4 independent experiments; 10 independent fields per animal were examined. Scale bars: 75 µm (**a**). Western blot analysis of HuC/D and S-100β expression on protein extracts obtained from the LMMP. β-actin was used as a loading control. Representative images of 3 independent experiments are provided (**b**). Protein signals of HuC/D and S-100β were determined through densitometric analysis of Western blots reported in (**b**). Data are reported as mean ± SD. number of mice for each experimental group = 3 mice per group. ^c^ denotes *p* < 0.05 vs. vehicle-treated mice. Light grey: vehicle; dark grey: abx; intermediate grey: wash-out (**c**).

**Figure 4 antibiotics-12-01000-f004:**
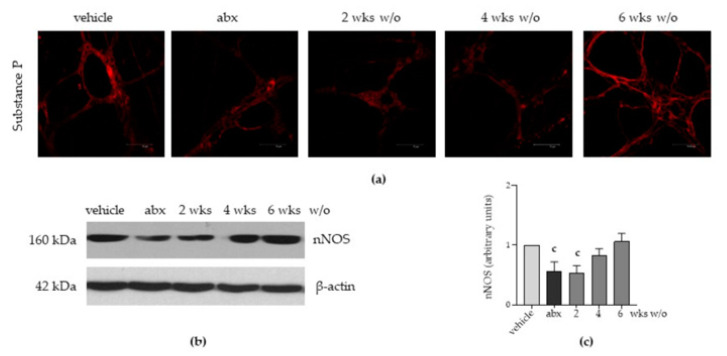
Mice were treated with vehicle or antibiotic mixture (abx) and sacrificed 2 weeks (wks) later; otherwise, mice were treated with abx and sacrificed 2, 4, or 6 weeks (wks) post antibiotic wash-out (w/o). Immunofluorescence on whole mount preparations of the distal ileum for substance P was performed. Representative images of 4 independent experiments; n = 3 mice per group; 10 independent fields per animal were examined. Scale bars: 75 µm (**a**). Western blot analysis of nNOS levels in total protein extracts obtained from the longitudinal muscle myenteric plexus (LMMP). β-actin was used as a loading control. Representative images of 3 independent experiments are shown (**b**). Protein signals of nNOS were determined through densitometric analysis of Western blotting reported in (**b**). Data are reported as mean ± SD. number of mice for each experimental group = 3 mice per group. ^c^ denotes *p* < 0.05 vs. vehicle-treated mice. Light grey: vehicle; dark grey: abx; intermediate grey: wash-out (**c**).

**Figure 5 antibiotics-12-01000-f005:**
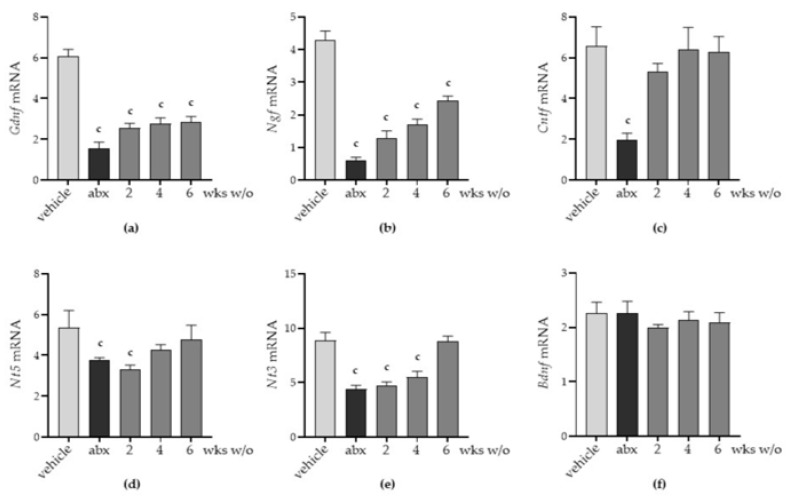
Mice were treated with vehicle or antibiotic mixture (abx) and sacrificed 2 weeks (wks) later; otherwise, mice were treated with abx and sacrificed 2, 4, or 6 weeks (wks) post antibiotic wash-out (w/o). The longitudinal muscle myenteric plexus (LMMP) was obtained at each time point, subjected to RNA extraction, and quantitative PCR of the neurotrophic factors: glial-derived neurotrophic factor (*Gdnf*) (**a**), nerve growth factor (*Ngf*) (**b**), ciliary neurotrophic factor (*Cntf*) (**c**), neurotrophin 5 (*Ntf5*) (**d**), neurotrophin 3 (*Ntf3*) (**e**), and brain-derived neurotrophic factor (*Bdnf*) (**f**). Light grey: vehicle; dark grey: abx; intermediate grey: wash-out. Data are reported as mean ± SD of the calculated fold changes over mRNA levels detected in vehicle-treated mice using the ΔC_T_ method. Number of mice for each experimental group = 6 mice per group. ^c^ denotes *p* < 0.05 vs. vehicle-treated mice.

**Figure 6 antibiotics-12-01000-f006:**
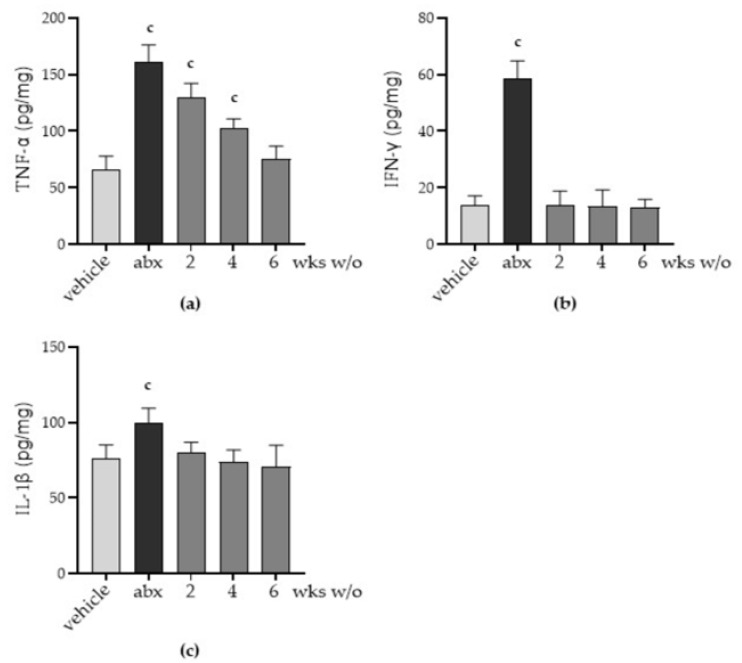
Mice were treated with vehicle or antibiotic mixture (abx) and sacrificed 2 weeks (wks) later; otherwise, mice were treated with abx and sacrificed 2, 4, or 6 weeks (wks) post antibiotic wash-out (w/o). The longitudinal muscle myenteric plexus (LMMP) was obtained at each time point. The pro-inflammatory cytokines tumour necrosis factor (TNF)-α (**a**), interferon (IFN)-γ (**b**), and interleukin (IL)-1β (**c**) were determined using commercially available ELISA kits. Light grey: vehicle; dark grey: abx; intermediate grey: wash-out. Data were normalized by total protein content measured using the bicinchoninic acid method and are reported as mean ± SD. Number of mice for each experimental group = 7 mice per group. ^c^ denotes *p* < 0.05 vs. vehicle-treated mice.

**Figure 7 antibiotics-12-01000-f007:**
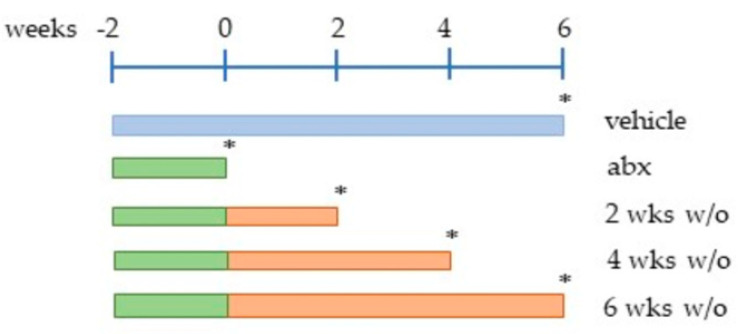
Experimental design of the study. Mice were administered a mixture of antibiotics every 12 h for 2 weeks (in green; wks, −2÷ 0). Animals were sacrificed at the end of the antibiotic treatment (abx) or 2, 4, or 6 weeks post antibiotic discontinuation (in orange; wash-out, w/o). Control animals (in blue; vehicle) received the same volume of the vehicle for the same period; since data obtained from the control animals at the different time points are comparable, the results of the controls are organized and reported as a single animal control group. Asterisks indicate the time of collection of the cecal content and the time of the sacrifice. Abbreviations: wks, weeks; abx, antibiotics; w/0, wash-out. * denotes sacrifice of the animals.

**Table 1 antibiotics-12-01000-t001:** Oligonucleotide sequences used for amplification of bacterial 16S and quantitative RT-PCR.

Target	Sequence
16S V2 region	Fw *: AGYGGCGIACGGGTGAGTAA
	Rv *: CYIACTGCTGCCTCCCGTAG
16S V6 region	Fw: AGGATTAGATACCCTGGTA
	Rv: CRRCACGAGCTGACGAC
*Gdnf*	Fw: TCAACTGGGGGTCTACG
	Rv: GCATCTGGGGGTCAACCT
*Ngf*	Fw: AGTTTTGGCCTGTGGTCGT
	Rv: GGACATTACGCTATGCACCTC
*Cntf*	Fw *: GGCCAAGCAAATGTAGCTCTT
	Rv *: GCCCCTGGGGAACTACTG
*Nt5*	Fw: CCCATCCAACATGACCCTA
	Rv: CAATGAGCTGCATGAGGAGA
*Nt3*	Fw: CGACGTCCCTGGAAATAGTC
	Rv: TGGACATCACCTTGTTCCACC
*Bdnf*	Fw: GAAGGCTGCAGGGGCATAGCAAA
	Rv: TACACAGGAAGTGTCTATCCTTATG
*Lif*	Fw: CGCCTAACATGACAGACTTCCCAT
	Rv: AGGCCCCTCATGACGTCTATAGTA
*Rn18s*	Fw: AACTTCTTAGAGGGACAAGTGG
	Rv: CGGACATCTAAGGGCATCAC

* Fw: forward; Rv: reverse.

**Table 2 antibiotics-12-01000-t002:** Antibodies used in the study.

Target (Host)	Clone	Source	Application
β-actin (mouse)	AC-15	Merck	WB
βIII-tubulin (rabbit)	polyclonal	Couvance	WM
HuC/D (mouse)	16A11	Abcam	WB
nNOS (rabbit)	polyclonal	Invitrogen	WB
Substance P (rabbit)	polyclonal	ImmunoStar	WM
S100β (rabbit)	EP1576Y	Millipore	WB
anti-mouse IgG	HRP conjugate	Merck	WB
anti-rabbit IgG	HRP conjugate	Merck	WB
anti-rabbit IgG	Alexa Fluor 555 conjugate	Invitrogen	WM

WB: Western blot; WM: whole mount.

## Data Availability

All data are reported in the Manuscript.

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
