# Peer review of "Antibiotic Treatment Induces Long-Lasting Effects on Gut Microbiota and the Enteric Nervous System in Mice"

_antibiotics, 2023, doi:10.3390/antibiotics12061000_

Round 1
Reviewer 1 Report
Dear Authors,
I have thoroughly read your work and I appreciate the depth of analysis you have provided on your research topic. Overall, I think your paper is well written and well-structured and I recommend it for publication. I only have a few minor questions. Perhaps you could consider including the answers to my questions in the discussion section if that seems appropriate to you.
1. How do you explain the persistence of effects after four weeks of antibiotic discontinuation?
2. Why specifically V2 and V6 regions of bacterial 16S genes were used and not other hypervariable regions?
3. Have you considered other factors that may have contributed to the observed effects on the enteric nervous system? Such as, for example, cytotoxicity of the antibiotic mix.
Looking forward to see your revised manuscript,
Sincerely,
Reviewer #
Author Response
Please, see the attached file.

Reviewer 2 Report
In this study, the authors assess the long-term impact of antibiotic-induced microbial dysbiosis on the structure and functions of the ENS in mice, and the impact of spontaneous recovery of the gut microbiota on gastrointestinal functions. This is an interesting study but some modifications are needed before publication to improve its quality.
Introduction
· L34: change “harmful” to “potentially harmful”
· L48-56: Do germ free mice show alterations in the development, structure and function of their ENS?
· L61-62 The authors state that “recovery does not mean retrieval of the same diversity and richness prior to antibiotic use”. However, the justification of this statement in the following sentences is not clear. Please improve the explanation.
Results
· In all figures, please replace the statistical symbols with letters in each of the 5 groups, in order to better visualize the significant differences between them. Values should be shown with SD, and not SEM.
· The authors state (L95-96) that the copy number in the bacterial genome partially recovered by the fourth week of wash-out, and only returned to the values reported in vehicle-treated mice after six weeks of antibiotic withdrawal. However, this statement does not coincide with the results shown in Figure 1a: no differences are in this parameter between 4 and 6 weeks, nor when comparing these values to those of the vehicle and the abx group. Please, also correct the sentence corresponding to these results in the Discussion section (L244-45, L249-50)
· In Figure 1b, OTU abundances at 4 and 6 weeks are not significantly different from the vehicle value, suggesting restauration of OTU abundance, contrarily to the authors’claim (L99-100).
· Please add the figure corresponding to beta-diversity of the microbiota for the different groups.
· The results for the post-abx period in Figure 3c are not described in the text.
· The meaning of the abbreviation LMMP was not given in the text earlier (L189)
· L189-93: the description the results is too superficial. What happens in the post-abx period? The differences are shown only with vehicle but it is not specified whether these values differ from the abx group. Is the restauration of these parameter absent, partial, total?
· Are the changes in gut microbiota correlated with the changes in the different parameters analyzed in the study?
Discussion
· L246: according to the authors (L121), no change in the structure of the ileum was observed in the study. Please correct the corresponding sentences in the text.
· L266: not only the innate immune system but also entero-endocrine system and the same ENS
· L269: please specify which trophic factors are referred to.
· L248 replace “prevalence” with “increased abundance”
· Some studies have shown that the antibiotic erythromycin directly inhibits nerve-mediated contractions by acting on enteric nerves and on longitudinal but not circular muscle (Minocha & Galligan, JPET 1991). Therefore, the possibility that the antibiotics used by the authors in this study may, in themselves, affect ENS independently of their effect on the microbiota, cannot be ruled out. This possibility should be discussed in the Discussion section.
The English lenguage needs some minor revision
Author Response
Please, see the attached file.

Round 2
Reviewer 2 Report
Please, add in the text a decription and interpretation of the results shown in Figure 1c (beta-diversity)
can be improved
Author Response
We thank Reviewer #2 for the comments on our Manuscript.
In the second round of revision we answered the comments. In particular, we added a description and interpretation of the results shown in Figure 1c in the text. Our comments are reported in lines 101-111 of the new version of the Manuscript. Moreover, we improved the English language in the Manuscript. A colleague fluent in English writing checked the new version of the Manuscript. All changes made in this second round of revision are marked up using the Track Changes function of Word.
We are looking forward to hearing from you.